# Efficient and Automatic Breast Cancer Early Diagnosis System Based on the Hierarchical Extreme Learning Machine

**DOI:** 10.3390/s23187772

**Published:** 2023-09-09

**Authors:** Songyang Lyu, Ray C. C. Cheung

**Affiliations:** Department of Electrical Engineering, City University of Hong Kong, Hong Kong; yang.lyu@my.cityu.edu.hk

**Keywords:** extreme learning machine, image processing, breast cancer, cancer diagnosis

## Abstract

Breast cancer is the leading type of cancer in women, causing nearly 600,000 deaths every year, globally. Although the tumors can be localized within the breast, they can spread to other body parts, causing more harm. Therefore, early diagnosis can help reduce the risks of this cancer. However, a breast cancer diagnosis is complicated, requiring biopsy by various methods, such as MRI, ultrasound, BI-RADS, or even needle aspiration and cytology with the suggestions of specialists. On certain occasions, such as body examinations of a large number of people, it is also a large workload to check the images. Therefore, in this work, we present an efficient and automatic diagnosis system based on the hierarchical extreme learning machine (H-ELM) for breast cancer ultrasound results with high efficiency and make a primary diagnosis of the images. To make it compatible to use, this system consists of PNG images and general medical software within the H-ELM framework, which is easily trained and applied. Furthermore, this system only requires ultrasound images on a small scale, of 28×28 pixels, reducing the resources and fulfilling the application with low-resolution images. The experimental results show that the system can achieve 86.13% in the classification of breast cancer based on ultrasound images from the public breast ultrasound images (BUSI) dataset, without other relative information and supervision, which is higher than the conventional deep learning methods on the same dataset. Moreover, the training time is highly reduced, to only 5.31 s, and consumes few resources. The experimental results indicate that this system could be helpful for precise and efficient early diagnosis of breast cancers with primary examination results.

## 1. Introduction

Breast cancer is one of the most common worldwide cancers, and it increasingly influences women. According to data from the WHO, in 2015 approximately 570,000 deaths from breast cancer were recorded, and over 1.5 million women (25% of all women with cancer) were diagnosed with breast cancer every year throughout the world [1]. Currently, new breast cancer cases of around 260,000 have been detected, and 44,000 deaths have been reported. Breast cancer is the most common among all the kinds of cancers [2]. There are various risk factors for developing breast cancer, such as sex, aging, estrogen, family history, gene mutations, and unhealthy lifestyle [3].

For the pathogenesis of breast cancer, the tumors usually start from ductal hyperproliferation and then develop into benign tumors, malignant tumors, or even metastatic carcinomas after constant stimulation by various carcinogenic factors [4]. Thus, breast cancer can commonly transfer to distant organs, such as the bone, lung, and even the brain, and it can increase the difficulty of treatment and curing. Fortunately, early diagnosis of breast cancer with certain examinations can lead to a good prognosis and maintain a high survival rate. In North America, the 5-year relative survival rate of breast cancer patients is above 80%, due to the timely detection of this disease [5]. Therefore, early and accurate detection with technology is increasingly important in breast cancer treatment. Medical imaging plays an essential role in the early diagnosis of breast cancer, in addition to the manual diagnosis of the professionals. There are several methods for diagnosis of breast cancer, and some of them can be selected according to doctor and patient. Mammography, which can be seen as a specialized X-ray for the breast, is a widely used screening approach in the detection of breast cancer, and it has been proven to help reduce mortality effectively [4]. Magnetic resonance imaging (MRI) can also be used for diagnosis, which is more sensitive but more expensive for some poor areas, considering breast cancer cases are found all over the world. Furthermore, ultrasound can also efficiently detect tumors in the breast, and ultrasound image (USI) is also important in diagnosis. Finally, biopsies—including different types, such as fine-needle aspiration, core biopsy, and open biopsy—can probe and obtain tissue, for further examinations. Among these methods, medical imaging can help detect organics and tumors efficiently, without harm to patients. However, image-based diagnosis is dependent on a series of factors. For example, the quality of medical images can be deteriorated by image noise [6]. In addition, in undeveloped areas, there may not be enough devices for the examinations of patients, nor enough radiologists for suggestions. To improve efficiency in the detection and diagnosis of breast cancer, an automatic and fast mechanism is required.

To realize an efficient and automatic system, computer-aided diagnosis (CAD) can serve as an assistant. CAD has become one of the major research subjects in medical imaging, and diagnostic radiology has become part of routine clinical work on the detection of breast cancer at many screening sites and hospitals [7]. With the help of CAD, radiologists or doctors can use the computer to analyze the image, and radiologists make the final decisions, which is different from simple computer diagnosis. Therefore, in clinical settings, radiologists can make decisions on a case-by-case basis. However, the quality of the image and the accuracy, sensitivity, and generality of the system will have a large influence on the diagnosis, and wrong results and analysis may even mislead the doctors and patients. Therefore, the diagnosis of breast cancer based on medical images can be simplified into the problems of segmentation, process, and classification of medical images by artificial intelligence (AI) methods, which can help the users to realize projects automatically with high efficiency.

However, diagnosis of cancer or any other illness is complicated, because many aspects need to be considered, such as other examination results, including blood pressure and electrocardiogram, genetic or family history, the feelings of the patients, and other information. Even the most experienced doctor or radiologist needs to be careful with every patient. Therefore, an efficient and automatic system can be useful in early or primary diagnosis, especially for a large number of patients, to give a primary result for the patients. For tools in relative applications, such as image processing, AI and machine learning models are often employed in different images, from large images, such as hyperspectral images [8,9,10], to cell images [11]. In the biomedical engineering area, these tools are helping to advance biomedical science in many ways, from improving image-based diagnostics to engineering strategies for improving movement related to injury, birth defects [12], or neurological or cardiovascular disease, such as detecting and diagnosing cancers [13,14], predicting behavior [15] and nerve responses to stimuli [16]. For the classification of the images of breast cancer, various algorithms have been used in different types of images, including MRI, ultrasound, mammography, and cytological images, with certain accuracy, based on different resources. The main steps of this process can be simply divided into two steps: training models with a training dataset and final decisions.

Some of the works on breast cancer classification are focused on whole-slide imaging (WSI) of needle cytology, especially those using conventional machine learning algorithms. Kowal et al. compared different algorithms on WSI, including k-nearest neighbors (k-NN) [17], when k = 7 was selected, the naive Bayes classifier with kernel density estimate [18,19,20], and decision trees [21], and the cases were classified as either benign or malignant on a dataset of 500 images, with a reported accuracy ranging from 96% to 100%. For other deep learning methods, George et al. [22] used the support vector machine (SVM), the probabilistic neural network (PNN), neural networks, and learning vector quantization (LVQ) on cytological images of breast cancer, with accuracy from 76% to 94%. S. Kaymak et al. [23] used artificial neural networks (ANN) on biopsy images of breast cancer and realized the classification with accuracy of 70.49% compared to the classification of the same dataset by backpropagation (BP), by which the accuracy was 59.02%. Michał Żejmo et al. [24] used needle cytology to realize the classification of breast tumors by convolutional neural networks (CNN). Among the different used architectures, the best patch classification accuracy reached 83%, obtained by GoogLeNet architecture. For deep learning methods, [25] proposed a novel deep learning algorithm, and the proposed framework’s different low-level features were extracted separately by three well-known CNN architectures: GoogLeNet, VGGNet, and ResNet. The combined features were fed into a fully connected layer, for the classification task. For the results, the separate GoogLeNet, VGGNet, and ResNet architecture individually provided average classification accuracies of 93.5%, 94.15%, and 94.35%, respectively, while the proposed framework provided accuracy of 97.525%. In addition to the combination of different models of CNN, Liu et al. showed the fully-connected-layer-first CNN (FCLF-CNN) method for breast cancer datasets [26]. This work suggested that a typical CNN may not maintain its performance for structured data. In order to take advantage of CNN, to improve the classification performance for structured data, the fully connected layers were embedded before the first convolutional layer, for the FCLF-CNN. After the test, the proposed CNN showed better performance than conventional methods, including CNN and pure multi-layered perceptrons (MLPs). T.S. Subashini et al. introduced the radial-basis-function neural network (RBFNN) for cytological patterns [27]. Compared to the SVM when processing the same dataset, the accuracy was 96.5664, which was better than 92.1316 for the SVM, and the sensitivity was 97.2953 compared to 86.7318, respectively. The cytology images did realize accurate examination results; however, this method can cause harm to patients and may require complex devices and systems. Currently, other detection methods can also be used for diagnosis, with no harm and high efficiency.

The mammogram was invented for detecting breast cancer and other diseases with high efficiency [28]. However, because of the complex structure of mammogram images, it is relatively difficult for doctors to identify breast cancer features [29]. Therefore, analyzing mammogram images requires a large computation amount or rich experience on the part of radiologists. For the works in this area, more deep learning methods have been proposed. W.M. Salama et al. [30] proposed that the combined deep learning methods of a pre-trained modified U-Net model and different deep learning models be utilized. InceptionV3, DenseNet121, ResNet50, VGG16, and MobileNetV2 were used, to build a framework of data augmentation + modified U-Net model + classifier networks. This work realized an accuracy of 99.43%, compared to other networks on the same dataset, whose accuracies ranged from 83% to 96%. H. Li et al. improved conventional DenseNet and generated DenseNet II for mammogram classification [29]. The accuracy of DenseNet II was 94.55%, and this was higher than the accuracy of conventional algorithms, including AlexNet, VGGNet, GoogLeNet, and DenseNet.

Ultrasound is widely used in clinical settings, because of its generality, with high quality for nearly all organics, and no harm or radiation to patients. Meanwhile, ultrasound is also a common and efficient examination method for breast cancer, especially in early diagnosis. For the analysis of ultrasound images, ref. [31] showed the method of the combination of multi-fractal dimension and ANN classifier, to recognize the ultrasound image representing positive (malignant tumors) or negative (normal and benign tumors), with the results manifested in high rates of precision (82.04%), sensitivity (79.39%), and specificity (84.75%). H. Feng et. al presented an improved vision transformer (ViT) model called ViT-Patch, which added a shared MLP head to the output of each patch token, to balance the feature learning on the class and patch tokens for the diagnosis of malignant tumors of breast cancer, with a best ACC of 89.8% and a SEN of 72.7% [32]. Y. Wang et al. presented a multiview convolutional neural network with transfer learning, and they achieved sensitivity of 0.886, specificity of 0.876, and a mean AUC value of 0.9468 with a standard deviation of 0.0164 in classification [33]. K. Jabeen et al. [34] used a probability-based optimal deep learning feature fusion method to classify the ultrasound image into benign, malignant, or normal. The breast ultrasound data were augmented and retrained by a DarkNet-53 deep learning model and were then processed in various steps. Ultimately, this method realized the best accuracy, of 99.1%. To summarize the results of AI applications for breast cancer images, a comparison of different methods is shown in Table 1.

From the review of related works above, deep learning methods and models including different kinds of CNN and revised CNN, are widely used in this area, because of the powerful learning ability and generality of the models, and they can be used in different kinds of images. However, these networks usually show their advantages in applications with big data, because a large amount of data are required in training and strengthening the performance, and they need to be updated frequently, consuming large resources on powerful computers and a long time in training. Moreover, the quality of the images also has a large influence on training, which may lead to wrong training and results in applications. Considering that the image classification system serves as an assistant to radiologists or doctors in clinical settings, and that it is usually used in early and primary diagnosis, a lightweight, flexible, and generalized system is required in applications. Therefore, we present a novel system based on a hierarchical extreme learning machine (H-ELM), to classify small-scale and general-format ultrasound images of the patient’s breast, showing the advantages in training time and resource utilization with good performance in accuracy. This system is easy to update with new datasets and diagnosis results, and it only requires general software in the biomedical engineering area and low-resolution images. According to our experiments on the selected dataset from the hospital, it can achieve higher accuracy, of 86.13%, and much less training time, of 5.31 s, compared to conventional deep learning methods.

The rest of this paper is organized as follows. Section 2 shows the instructions for the ultrasound image dataset. Section 3 presents the design and each part of the system. Section 4 shows the experiment results and discussion, and the conclusion and future research direction are shown in Section 5.

## 2. Materials

In general, to build an automatic diagnosis system, a dataset with correct labels under supervision is essential. In this work, we used the breast ultrasound images (BUSI) dataset [35] for training and testing. The data were collected from 600 women, in ages ranging from 25 to 75 years old, in 2018. The devices used in the scanning process were the LOGIQ E9 ultrasound system and the LOGIQ E9 Agile ultrasound system, which are commonly used worldwide, and the data had been previously stored in a DICOM format in the Baheya Hospital of Egypt, with a resolution of 1280×1024. The examination results were classified into three types: normal, benign, and malignant. The original ultrasound images from the patients are shown in Figure 1. After pre-processing and transformation of the format, the images were transformed from the clinically specified DICOM format into the more common PNG format, and the size of the images was approximately 600×600 pixels, occupying approximately 350 kb in storage. Because the images were general to use and on a small scale, they were very suitable for H-ELM fast training for a lightweight model in applications.

## 3. Methods

### 3.1. Overview

In this work, we aimed to build an automatic and efficient system for clinical images of breast cancer, and the design flow is shown in Figure 2. The total design can be divided into 4 modules: pre-processing of data, training the models, results for diagnosis, and manual diagnosis for final decision. To achieve the fast training speed and reduce time consumption, an ultrasound image is selected, since it is a popular choice because it is well tolerated by patients, widely available, does not require intravenous contrast or ionizing radiation, and is relatively inexpensive [36], compared to mammography, MRI, and biopsy, which may need more expense or harm and are more complicated in collecting. To realize fast training with fewer resources, each pixel of the ultrasound image is processed from gray-gradient into logic and keeps the essential shape and scale of the tumors at the same time. Then, the image is segmented into smaller sizes of 28×28 pixels on Matlab, to further reduce the scale in processing.

To build the system, a supervised model with high efficiency is required, for learning the diagnosis standard from the dataset, and it should be suitable for 2D images. Therefore, we use the hierarchical extreme learning machine for training and one more extreme learning machine for the final decision, because the extreme learning machine has shown good performance in face classification [37], diagnosis of lung cancer [38], brain–machine interface (BMI) experiments on mice [39], and EEG signal classification [40]. For this project, more layers were added, for better performance in accuracy and a more stable structure in calculations. Therefore, the H-ELM and the single-layer ELM served as the framework for learning and classifying ultrasound images. And, in this project, we also tested and selected the most suitable scale of hidden layers for the best performance in accuracy and training time.

For this dataset and the labels, the framework outputted 3 results: benign, malignant, and normal without any tumors, and the single-layer ELM made the final decision in the models. For the test or automatic diagnosis module, the framework will decide which kind of tumor it is, according to the training dataset. Therefore, the testing results depend on the confidence of this system, and certain training datasets should be well learned by a suitable number of hidden layers.

Finally, this system is designed as an early diagnosis automatic system for breast cancer, aiming to improve the accuracy of diagnosis without the supervision of professional radiologists or doctors. Currently, not any system or examination can replace the role of the doctor in clinical settings; therefore, the results from the system still need to be transferred into suggestions for the professionals, and doctors can make the final diagnosis, while also considering information such as medical and genetic history, clinical manifestation, and other examination results.

### 3.2. Processing of Images

Before the diagnosis, the ultrasound images from different devices and patients should be generalized and processed before being sent to the system. In this work, we present a method of pre-processing images with general software, and the total process is shown in Figure 3. This work is designed for general usage in the early diagnosis of breast cancer, so the pre-process steps only require public software and a common image format of PNG. In practice, the format of the images collected from the ultrasound devices is DICOM, which may not be generally used by other systems or software. Thus, for the ultrasound breast cancer dataset, RadiAnt is used, which is a common software in medical and clinical settings, for transforming the format of images from DICOM into PNG format. Then, another software, named Fast Photo Crop, is used, to obtain ultrasound images with similar sizes, of 600×600 pixels, or this part can be replaced by other tools to generalize the images, such as Matlab. Moreover, to improve the efficiency and accuracy of training and processing, the images are segmented further, because in the biomedical engineering area, image segmentation is the action of grouping pixels according to predefined criteria, in order to build regions or classes of pixels [41]. Therefore, considering the powerful learning abilities of the H-ELM, the images are further simplified from gray images into logic images, and reduce the sizes one more time. Here, ground truth (image boundary) is performed, to make the ultrasound dataset beneficial on Matlab, an easy segmentation is established, and the images are finally processed into binary images of 28×28 pixels. The samples of mask images and simplified images are shown in Figure 4. After the process, the features of the tumors in the ultrasound images—such as the shape, location, and size—are also centralized, to some degree, which also helps to maintain accuracy in classification.

### 3.3. ELM Theory

In 2006, G. Huang et al. present the theory, algorithm, and applications of extreme learning machines [42], which help improve the training speed of slow gradient-based learning algorithms, and avoid all the parameters of the networks tuned iteratively. The theory of ELMs shows a new method in training and learning, compared to conventional machine learning algorithms. For the main process of an ELM, firstly, the hidden layers are set randomly or according to the previous test results. Then, the algorithm will calculate and update the hidden layers by iterative calculations, which requires a large amount of time and resources in this step. Finally, after long training, the framework can be used in applications. However, as the applications may not have enough time and resources, and they need to be realized in a short time, conventional methods, such as back-propagation (BP), may not meet these requirements. However, with a simple single-layer ELM, the framework may not have enough ability to learn with high accuracy in results, even if the size of the hidden layer is very large [43]. The ELM also involves inversion calculation of matrices; a large hidden layer may cause unstable results. Therefore, for applications with a more complicated dataset, the H-ELM is used with higher accuracy and more flexibility, and it is easier to set parameters.

To build a stable and efficient H-ELM, several single-layer ELMs are used and connected one by one, and the structure of a single-layer ELM is shown in Figure 5. For the general single-layer ELM calculation process, the training set is shown in *X*: a set of *N* labeled pairs $\left(x_{i},y_{i}\right)$, where $x_{i}\in R$ is the *i*th input vector and $y_{i}\in R$ is the associate expected “target” value. Then, to calculate the output, the function f(x) of an ELM is written as
(1)$$f\left(x\right)=\sum_{i=1}^{M}w_{j}▪a\left(r_{j}▪x+b_{j}\right).$$

The input layer, considered with *M* neurons, is connected to the hidden layer with *H* neurons through a set of weights, $r_{j}\in R,j=1,\ldots ,H$, while the *j*th hidden neuron embeds a bias term $b_{j}$ and a nonlinear activation function of a(▪). A vector of weighted links w∈R calculates the output neuron with the hidden layer. The quantities $r_{j},b_{j}$ in (1) are set randomly and not subject to any optimization. Let *H* denote the activation matrix, where $h_{ij}\in H\left(i=1,\ldots ,N;j=1,\ldots ,H\right)$ is the activation value of the *j*th hidden neuron for the *i*th input pattern $h_{ij}=a\left(r_{j}▪x_{i}+b_{j}\right)$. Detailed results are shown in (2):(2)$$H\left(w_{1},...,w_{\tilde{n}},b_{1},...,b_{\tilde{n}},x_{1},...,x_{\tilde{n}}\right)=\left[\begin{array}{ccc} a(w_{i}{\dot{x}}_{j}+b_{i}) & \cdots & a(w_{\tilde{N}}\dot{x_{1}}+b_{\tilde{N}})\\ \vdots & \cdots & \vdots \\ a(w_{i}{\dot{x}}_{N}+b_{i}) & \cdots & a(w_{\tilde{N}}\dot{x_{N}}+b_{\tilde{N}}) \end{array}\right]$$

Overall, the training of the ELM will be finished with minimization of the convex cost:(3)$$\min_{\left(w,b\right)}{∥\begin{array}{c} Hw-Y \end{array}∥}^{2}.$$To summarize, a straightforward and general procedure to train an ELM consists of the following steps: First, randomly set the parameters $r_{j},b_{j}$ for each hidden neuron, where j=1,…,H. Second, compute the activation matrix *H*. Third, compute the output weights, by solving a pseudo-inverse problem in Equation (Equation 3). The core computation is shown in Equation (Equation 4):(4)$$\min_{\left(w,b\right)}{∥\begin{array}{c} Hw-Y \end{array}∥}^{2}+\lambda {∥\begin{array}{c} w \end{array}∥}^{2}.$$

### 3.4. H-ELM Theory

To improve the performance of ELMs in simplifying structure and avoiding unstable matrices and low accuracy, the H-ELM was selected for this project. The H-ELM, presented in 2015 by J. Tang et al. [43], is a multi-layer neuron network based on the theory of the ELM with high efficiency and accuracy. It mainly can be divided into two parts: an unsupervised multi-layer ELM and supervised feature classification. For the first part, there is an ELM sparse auto-encoder [44], utilized for feature extraction and representation. For the second part, the obtained features from the first part are scattered by a randomly generated matrix, and then an original single-layer ELM works for the final decision-making [40]. A general framework of the H-ELM is shown in Figure 6. Generally, the multilayer ELM is built by single-layer ELMs with a connection to one other. The output and weights of the last layer will be the input data for the next layer. Then, an N-layer unsupervised multilayer ELM is performed, to eventually obtain the high-level sparse features. The connection of each hidden layer can be represented as
(5)$$H_{i}=g\left(H_{i-1}\dot{\beta }\right).$$ In this equation, $H_{i}$ is the output of the *i*th layer (i∈[1,K]), according to the connection, so $H_{i-1}$ is the output of the i−1th layer. Among the overall structure, g(·) represents the activation function of the hidden layers, and β represents the output weights. Each layer not only works as a part of the training in a H-ELM, but also as an independent module at the same time. Therefore, when it is used for classification or decision making, it is randomly set and works for calculating the results, respectively. This shows the high flexibility of each layer of ELM in applications.

To summarize the whole process, it can be separated as follows:Step 1: Given a training set,
(6)$$N=\left(x_{i},t_{i}\right)|x_{i}\in R^{n},t_{i}\in R^{m},i=1,...,N,$$
activation function g(x) and hidden neuron number $\hat{N}$;Step 2: Calculate the hidden layer output of the last layer by Equation (Equation 4);Step 3: Calculate the output weight of the last layer β;Step 4: Connect each layer with a certain sequence through $H_{i}=g\left(H_{i-1}\dot{\beta }\right)$;Step 5: Make the final decision on the original ELM, with the help of the auto-encoder.

In this work, a 4-layer H-ELM and a single-layer ELM were used for training and for the final decision, and realized good performance in the classification of 3 types of images. The test and selection of different numbers of hidden layers and the performance results are shown in the next section.

## 4. Experiment Results and Discussion

In this work, all of the experiments including training and tests were performed on a PC with Intel(R) Core(TM) i5-6700 CPU @2.70 GHz, 24 GB 2133 MHz RAM hardware and Matlab 2020a software. A public dataset of women aged between 25 and 75 years old, which was collected in 2018 by Baheya Hospital, Egypt, was used for training and testing. After pre-processing of the dataset, a total of 798 samples were divided into three types: benign, malignant, and normal. Nearly 80% of the samples were used for training, and the rest were used for testing. We selected ultrasound images that were 28×28 pixels in size for the early diagnosis. Not only did the small-size images require less training time, but also they worked as feature extraction, to some degree. Under the same H-ELM framework, different sizes of images were used for the test, and the results are shown in Table 2. From the table, for the processed ultrasound images, more pixels could not offer more useful information for the framework, and the accuracy result was 4% lower than the results of fewer pixels. However, the training time of large-scale images increases by nearly 119× compared to small-scale ones. Therefore, images of 28×28 pixels were used for this design and, in applications and clinics, this design can also realize early diagnosis, as assistance to radiologists and doctors.

Generally, for the algorithms in machine learning, the increase of the scale of the hidden nodes may improve the learning ability and may require more training time for the same dataset. However, for a single-layer ELM, a large-scale hidden layer may cause unstable matrices in the inversion calculation, and the calculation results may have an influence on the accuracy of the final decision. Although the H-ELM reduces these problems, by separating the hidden layer into more layers, the scale of each layer still has an effect on the accuracy, and the result may approach a limitation, even if the layer is very large. Therefore, the suitable scale of the hidden layer can obtain the best results with a short training time. For this dataset, with the increase in the number of hidden layers, the training time increased exponentially with the testing of hidden nodes in each layer, from 300 to 5000, as shown in Figure 7, and the number of hidden nodes in each layer was selected and tested based on the H-ELM used for similar works [43]. However, for the accuracy of the test in this work, it did not increase with more numbers of hidden nodes. As shown in Figure 7a, the accuracy increased rapidly with the increase of hidden nodes in each layer from 300 to 800, and the accuracy arrived at a peak of 82.02%. But it fluctuated with more hidden nodes: for example, the accuracy of 2000 hidden nodes was lower than that of 800. Finally, during the total testing, the three-layer H-ELM realized the highest average accuracy, with 3000 hidden layers of 82.48%. According to the theory and applications of ELMs, a very large scale of the hidden layer may not fulfill the requirement of learning ability [43], and there are untrusted results, because in the test the matrices in calculation may near singular value. Therefore, more layers are needed, to realize better performance in accuracy with similar training times.

Based on the experiments, ultimately, a four-layer H-ELM was used for this project, and the results are shown in Figure 7b. The accuracy on the same scale of the hidden layer increased, and the highest accuracy was 86.13% when there were 1000 hidden nodes in each layer. The highest accuracy realized an improvement of 4% compared to the three-layer, and the training time only saw a slight increase, relatively. Because a large number of hidden nodes still had an influence on accuracy, the peak of accuracy in Figure 7b was realized with 1000 hidden nodes and 5.31 s in training, and it decreased after the peak.

To illustrate the trend and the peak under different layers, and to find the most suitable number of hidden nodes, we selected three test results with high accuracy under a certain number of hidden nodes. The results are shown in Table 3. From the table, we can see that the four-layer H-ELM with 1000 hidden nodes of each layer had the best accuracy performance, with only 5.31s training time.

Compared to other deep learning methods for the same dataset, classification based on the H-ELM for reduced-size images without other feature extraction methods also shows advantages, in terms of accuracy. For details, the comparison of accuracy is collected in Table 4. In the bio-medical engineering area, many types of deep learning methods are trained and optimized for different applications, because the learning in this area is also improving, based on more experience and better devices with higher precision and sensitivity. Therefore, the models or the frameworks need to be updated, according to the new dataset or cases from the patients or experiments, and more frequent updates can also help to improve accuracy in clinics and applications. In addition to the accuracy of testing, training time also plays an important role in the whole project. Moreover, the method proposed in this work requires more general and simple ways to process the images, which are more friendly for any users without any professional experience or knowledge. Instead of other feature extraction or data cleaning methods, the ultrasound images in this method do not need complex processing, and only use small-scale images, which require fewer resources to store, and general devices can fulfill the usage in early diagnosis. Although it may reduce the accuracy to some degree, the frequent update of the training dataset with fast training speed can also help to keep the model learning the latest images of examination, which also improves stability and reliability in applications.

## 5. Conclusions

A novel system for the early diagnosis of breast cancer, based on the hierarchical extreme learning machine, was introduced, with fast training time, high accuracy, and low resources, and this method directly processed and classified the images of only 28×28-pixel ultrasound images. We applied a H-ELM with a suitable number of layers and scales for the dataset after adjustment, and we realized high efficiency and accuracy, compared to conventional deep learning algorithms. After the analysis by the system, the results of the ultrasound images would be classified into three types: benign, malignant, and normal, with an accuracy of 86.13% and a training time of 5.31 s, and the results can be used by professional radiologists and doctors for further diagnosis with other information about the patient. Although this early diagnosis system cannot replace the professional entirely, since there are more aspects, such as blood tests and genetic history, this system shows high efficiency in training, classifying, and accuracy. It can be updated frequently with new datasets and diagnosis results, to improve the accuracy, with more information involved in the system. At the same time, this system only needs ultrasound images of the breast, so that it can be used in a large number of images in specific circumstances, such as the body examination of students, workers, and employees. For further improvement of the system, fast and adaptive feature extraction algorithms will be implemented, to realize higher accuracy, and the features of the H-ELM also show its potential on hardware, such as the low resources occupation and simple structures. Therefore, hardware implementation or training will be the next direction in our further research.

## Figures and Tables

**Figure 1 sensors-23-07772-f001:**
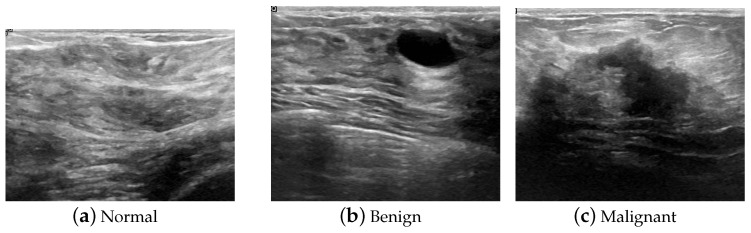
Samples of ultrasound images of BUSI dataset.

**Figure 2 sensors-23-07772-f002:**
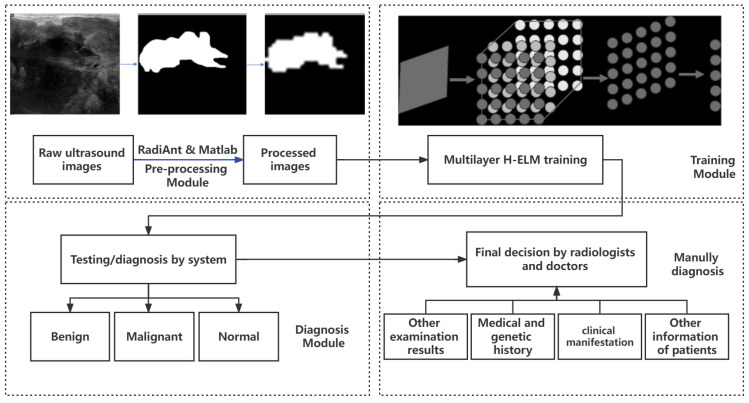
Design flow of the system.

**Figure 3 sensors-23-07772-f003:**
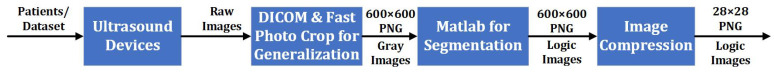
Design flow of the image pre-processing.

**Figure 4 sensors-23-07772-f004:**
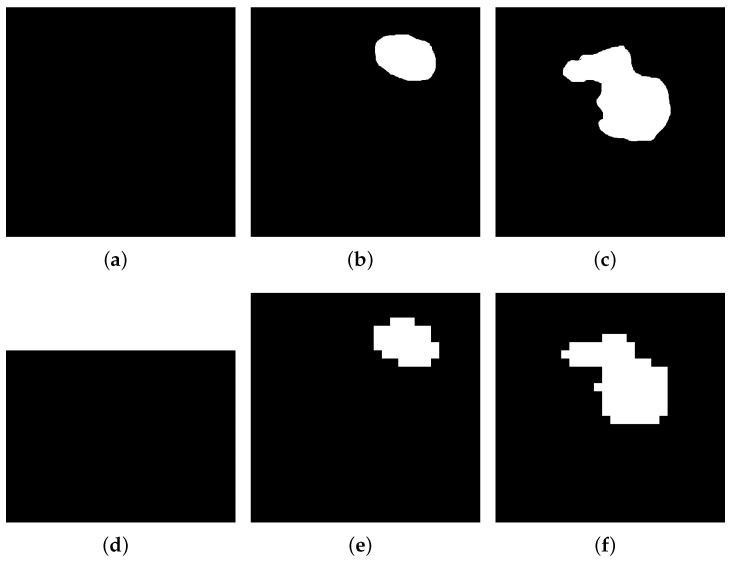
Samples of different sizes from pre-processed images of BUSI dataset: (**a**) normal image of 600×600 pixels; (**b**) benign image of of 600×600 pixels; (**c**) malignant image of of 600×600 pixels; (**d**) normal image of of 28×28 pixels; (**e**) benign image of of 28×28 pixels; (**f**) malignant image of 28×28 pixels.

**Figure 5 sensors-23-07772-f005:**
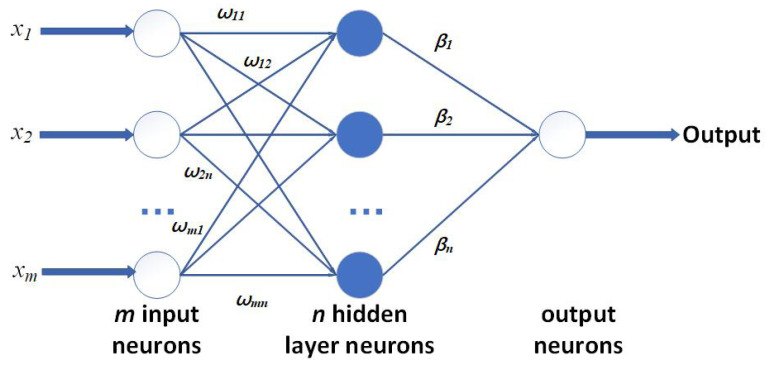
Basic structure of single-layer ELM.

**Figure 6 sensors-23-07772-f006:**
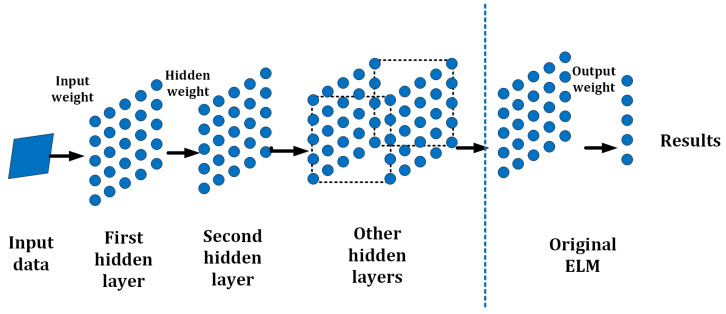
Basic Structure of the H-ELM.

**Figure 7 sensors-23-07772-f007:**
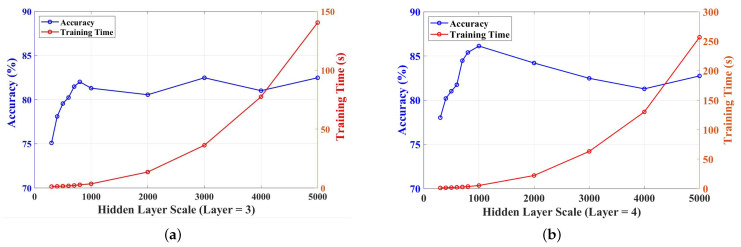
Accuracy and training time under different numbers of layers and hidden nodes: (**a**) trend of different scales of three-layer H-ELMs in accuracy and training time; (**b**) trend of different scales of four-layer H-ELMs in accuracy and training time.

**Table 1 sensors-23-07772-t001:** Comparison of different methods.

Authors	Method	Accuracy/AUC
W.M. Salama et al. [30]	Combined deep learning methods	99.43%
H. Li et al. [29]	DenseNet II	94.55%
H. Feng et al. [32]	ViT-Patch	0.898
Y. Wang et al. [33]	Multiview convolutional neural network	0.9468
M.A. Mohammed et al. [31]	Combination of multi-fractal dimension and ANN classifier	82.04%
K. Jabeen et al. [34]	Probability-based optimal deep learning feature fusion method	99.1%

**Table 2 sensors-23-07772-t002:** Different sizes of images processed by the same framework (the H-ELM).

Size of Image	Number of Layer	Number of Hidden Nodes in Each Layer	Test Accuracy	Training Time
28×28	3	700	79.56%	2.12 s
600×600	3	700	76.35%	253.57 s

**Table 3 sensors-23-07772-t003:** Performance with high accuracy, with different numbers of hidden layers and hidden nodes.

Layers	Number of Hidden Nodes	Accuracy	Training Time
3	800	82.02%	2.67 s
3	3000	82.48%	36.27 s
3	5000	82.48%	140.51 s
4	700	84.48%	2.85 s
4	800	85.40%	3.62 s
4	1000	86.13%	5.31 s

**Table 4 sensors-23-07772-t004:** Comparison of different methods in accuracy.

Method	Accuracy
Inception V3	74%
InceptionResnet V2	78%
Densenet	70%
Mobilenets	51%
VGG	73%
Improved-InceptionV3 [45]	80%
InceptionResNetV2	82.93%
GoogleNet	74.67%
SqueezeNet [46]	77.33%
The proposed method (H-ELM)	86.13%

## Data Availability

This paper uses the public dataset of breast ultrasound images, published in [35].

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
