# Peer review of "Efficient and Automatic Breast Cancer Early Diagnosis System Based on the Hierarchical Extreme Learning Machine"

_sensors, 2023, doi:10.3390/s23187772_

Round 1
Reviewer 1 Report
The proposed approach provides novelty in contribution and methodology. But, in this review round, major revision in terms of technical details is needed. Also, paper organization can be improved. In this respect, some comments are suggested to describe technical details.
1. Your proposed approach is compared with deep learning-based methods in this scope. So it is suggested to discuss about runtime of your proposed approach briefly (compare with other methods in terms of runtime is not needed)
2. Did you use a specific train-set to evaluate compared deep networks in the Table 3 (DenseNet, InceptionV3, etc?
3. What is the meaning of “[? ]” in the Page 7, line 256?
4. How do you select the number of hidden nodes in MLP? Did you evaluate based on different values?
5. MLP and ELM are used widely for medical cancer detection. For example, I find two papers titled “Identifying malignant breast ultrasound images using ViT-patch”, and titled “Developing a Tuned Three-Layer Perceptron Fed with Trained Deep Convolutional Neural Networks for Cervical Cancer Diagnosis”, which has enough relation. Cite these papers and some other as related works.
6. Pre-processing phase should be described in a clear way with technical details. Exactly, which technique did you use for tumor segmentation?
Reviewer 2 Report
Authors present an efficient and automatic diagnosis system based on the Hierarchical Extreme Learning Machine (H-ELM) for breast cancer ultrasound results with high efficiency. Furthermore, this system only requires ultrasound images on a small scale of 28 × 28 pixels, which requires fewer resources and more general devices.
-The abstract part needs to explain the writing motivation and research process of the article more clearly
-The parameter definition of the formula needs to be clearer
-The conclusion part needs to explain the research limitations of the paper.
-More statistical methods are recommended to analyze the experimental results.
-"Innovation" and "the operational success" of work must be improved in Section 1.
- How to tune the control parameters of Section 4 should be explained; Then the effect of their change in a wide range on the performance of the problem(s) should be investigated and reported.
-In order to highlight the introduction, some latest references should be added to the paper for improving the reviews part and the connection with the literature. For example, https://doi.org/10.3390/rs15133402;
http://dx.doi.org/10.1145/3513263;
https://doi.org/10.3389/fendo.2022.974063 and so on.
- What is the practical difficulty of the research in the paper? And can this article be further studied?
Extensive editing of English language required
Reviewer 3 Report
The aim of the proposed work is interesting in more ways than one, but the authors miss the mark by proposing a purely methodological approach without considering its practical application.
Indeed, to make an efficient diagnosis, we also need to know the evolution of the disease, its nature (by biopsy) and its location (by segmentation: the authors can draw inspiration from the key reference published in Cancers and cite it: https://doi.org/10.3390/cancers14184399).
On a practical level, as a hospital practitioner, it is almost impossible to make a decision based on this image.
The authors should not lose sight of the fact that their work is intended to be useful in routine hospital practice, and not as an application of known algorithms to a database.
Acceptable English
Round 2
Reviewer 1 Report
Most of comments has been considered by authors in the revised version. The revised version is better than original submission in terms of technical details and paper organisation.
Reviewer 2 Report
This paper can be accepted now.